# Ultraviolet irradiation-responsive dynamic ultralong organic phosphorescence in polymeric systems

Yongfeng Zhang[1], Liang Gao[1], Xian Zheng[1], Zhonghao Wang[1], Chaolong Yang [1✉], Hailong Tang [1], Lunjun Qu[1], Youbing Li[1] & Yanli Zhao [2✉]

Room temperature phosphorescence (RTP) has drawn extensive attention in recent years. Efficient stimulus-responsive phosphorescent organic materials are attractive, but are extremely rare because of unclear design principles and intrinsically spin-forbidden inter-system crossing. Herein, we present a feasible and facile strategy to achieve ultraviolet irradiation-responsive ultralong RTP (IRRTP) of some simple organic phosphors by doping into amorphous poly(vinyl alcohol) matrix. In addition to the observed green and yellow afterglow emission with distinct irradiation-enhanced phosphorescence, the phosphorescence lifetime can be tuned by varying the irradiation period of 254 nm light. Significantly, the dynamic phosphorescence lifetime could be increased 14.3 folds from 58.03 ms to 828.81 ms in one of the obtained hybrid films after irradiation for 45 min under ambient conditions. As such, the application in polychromatic screen printing and multilevel information encryption is demonstrated. The extraordinary IRRTP in the amorphous state endows these systems with a highly promising potential for smart flexible luminescent materials and sensors with dynamically controlled phosphorescence.

[1] School of Materials Science and Engineering, Chongqing University of Technology, Chongqing, China. [2] Division of Chemistry and Biological Chemistry, School of Physical and Mathematical Sciences, Nanyang Technological University, Singapore, Singapore. ✉email: yclzjun@163.com; zhaoyanli@ntu.edu.sg

Intelligent stimulus-responsive polymer materials, whose physical properties can be controlled by external stimuli such as light, pH, temperature, and pressure, serve a broad range of applications[1–4]. Light is an attractive stimulus for constructing responsive nanosystems, and ultraviolet (UV) light-responsive polymeric nanomedicine has received a lot of attention for their applications in on-demand and spatiotemporal drug delivery or disease therapy[5,6]. However, a significant challenge inherent to most stimulus-responsive organic systems is poor long-term stability and durability. Although a substantial research progress has been made in the field, developing more stable and advanced responsive systems for a series of applications is still highly desired[7]. Since room temperature phosphorescence (RTP) has been reported at first under inert conditions in the 1960s[8], their photonic and electronic properties which involve triplet excited states show promising potential for applications in next-generation technologies[9,10]. As an example, ultralong RTP is essential for state-of-the-art information security and biological applications. Noticeably, the absorption band of the vast majority of RTP molecules lies in the near UV spectral range. It is therefore expected that RTP materials are less prone to instability under UV irradiation when compared to other light-responsive systems. Therefore, the exploration of stimulus-responsive polymer-based RTP materials is attractive and important.

The intrinsic disadvantages of inorganic phosphorescence materials, including the harsh preparation conditions, the scarcity of rare metal resources, and high toxicity, have led to the search for metal-free alternatives[11]. Good electroconductivity and a myriad of molecular architectures paired with good biocompatibility and low cost for metal-free organic systems fueled a noticeable research trend to transfer concepts from inorganic phosphorescent materials to organic ones[12,13]. Organic phosphorescence materials have drawn vast interests for promising technological applications[14–17].

In order to enhance organic phosphorescence under ambient conditions, several challenges, such as promoting the intersystem crossing (ISC) process or suppressing nonradiative transitions ($k_{nr}$), need to be addressed. Until now, a set of feasible strategies has been proposed to obtain ultralong phosphorescence at room temperature, including host–guest interactions[18], halogen bonding[19], crystallization[20], and molecular packing[21]. Although the construction of a rigid environment via crystal engineering is a common method to achieve long-lived phosphorescence emission at room temperature[22], crystal-based RTP materials often suffer under poor flexibility, reproducibility, and processability, hampering their practical applications in cases where flexible, processable, and stretchable response systems are needed. To solve these challenges of crystal-based RTP materials, organic polymeric materials which are capable of emitting ultralong phosphorescence at room temperature were developed through homopolymerization[23,24], radical binary copolymerization[25,26], or loading small molecules into rigid polymer matrices[27–30]. Nevertheless, because of inefficient spin–orbit coupling (SOC), susceptibility of triplet excitons, and impurity quenching, organic phosphors usually exhibit diminished phosphorescence under ambient conditions. Even more challenging is the development of long-lived RTP materials with stimulus-responsive properties.

Many aspects of stimulus-responsive RTP materials have been studied, such as mechanoluminescence[31], excitation-dependent color-tunable phosphorescence[26–28], time-dependent afterglow[32], and temperature-activated phosphorescence[33]. However, intelligent stimulus-responsive phosphorescence materials have not been well investigated[34]. In particular, it is necessary to develop the irradiation-responsive ultralong RTP (IRRTP) systems. Modulation of long-lived triplet excitons to improve the efficiency of RTP is difficult, because of the numerous complex and competitive decay channels (Fig. 1a)[9,35–42], such as slow rate constant

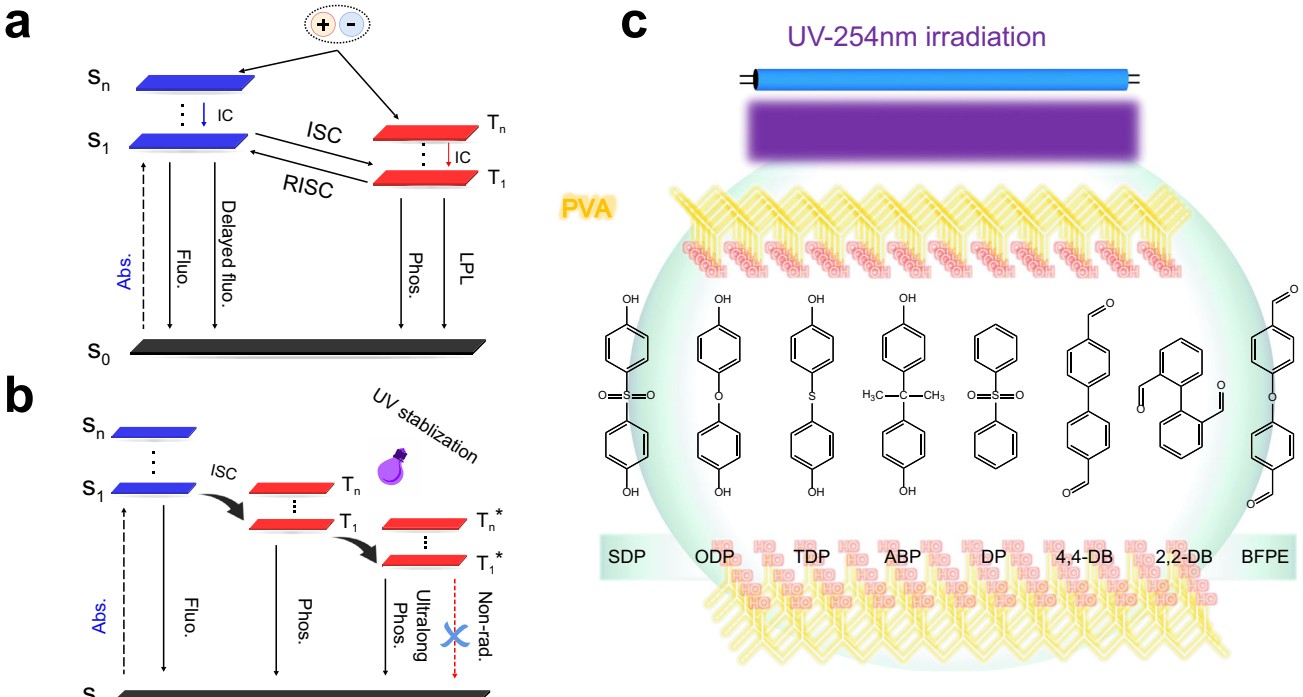

**Fig. 1 Schematic representation for the manipulation of an UV irradiation-responsive organic molecule-doped PVA system. a** Schematic illustration the competitive decay channels of luminescent materials. **b** Proposed mechanism and **c** molecular structures for irradiation-responsive ultralong RTP systems. Singlet excitons ($S_n$) produce triplet excitons ($T_n$) through the ISC process, and suppressing nonradiative transitions of the molecules leads to efficient phosphorescence. The formation of a rigid surrounding matrix can stabilize the triplet excitons and generate radiation-dependent phosphorescence. Abs. absorbance, Fluo. fluorescence, Phos. phosphorescence, LPL long persistent luminescence, IC internal conversion, Non-rad. nonradiation.

of phosphorescence ($k_p$) competing with fast nonradiative transition ($k_{nr}$) and quenching rate ($k_q$), strong SOC effect of function groups promoting ISC, triplet–triplet energy transfer, and reverse intersystem crossing (RISC) from triplet to singlet excited state. As a solution, the incorporation of phosphors into a rigid matrix may suppress nonradiative deactivation pathways. Recently, RTP materials based on introduction of phosphors into polymer backbone chains as well as based on doping or embedding phosphors in a rigid polymer matrix have been developed[29,40,43]. Thereby, most of the existing polymeric RTP materials benefit from different interactions between the phosphor and the polymer matrix, such as hydrogen bonding and ionic bonding. Inspired by this approach, we incorporate some simple phosphors into a poly(vinyl alcohol) (PVA) matrix to develop stimulus-responsive polymeric RTP systems.

Herein, we report IRRTP of eight pure organic phosphors (4,4′-sulfonyldiphenol (SDP), 4,4′-oxydiphenol (ODP), 4,4′-thiodiphenol (TDP), bisphenol A (ABP), diphenyl sulfone (DP), (1,1′-biphenyl)-4,4′-dicarboxaldehyde (4,4-DB), (1,1′-biphenyl)-2,2′-dicarboxaldehyde (2,2-DB), and 4,4′-oxydibenzaldehyde (BFPE)) by doping into the PVA matrix. The functional groups of these phosphors could form abundant hydrogen bonding interactions or enable a strong SOC effect to promote ISC. Meanwhile, the nonradiative decay could be efficiently suppressed by multiple hydrogen bonding interactions, and could be further inhibited by covalent bond (C–O–C) formation after 45 min irradiation under 254 nm UV-light. At the same time, the C–SO$_2$–C, C–O–C, C–S–C, and C–C(CH$_3$)$_2$–C groups between respective benzene rings in SDP, ODP, TDP, and ABP show different degrees of steric effect. For example, the sulfone group (O=S=O) in SDP can enable a strong SOC effect to promote the ISC, while the larger C–C(CH$_3$)$_2$–C group in ABP would lead to an obvious steric effect. Hybrid films consisting of these phosphors in the PVA matrix exhibit obvious irradiation enhanced RTP emission. Surprisingly, irradiation-induced ultralong green and yellow phosphorescence by doping SDP and 2,2-DB into the PVA matrix was observed, respectively. Meanwhile, the dynamic phosphorescence lifetime of the SDP-doped film could be tuned from 58.03 to 828.81 ms after irradiation under ambient conditions, representing a very rare phosphorescence-enhanced emission. Due to the suppression of nonradiative transitions by combining hydrogen bonding with cross-linking under UV irradiation, the triplet excitons for IRRTP in these doped polymer systems could be successfully stabilized (Fig. 1b). Thus, our approach offers a general design principle to develop UV irradiation-stimulated ultralong phosphorescence in amorphous polymers under ambient conditions.

## Results

**Photophysical properties of irradiation-enhanced phosphorescence films.** Numerous hydrogen bonding interactions among the PVA chains form a relatively rigid polymer microenvironment, which not only restrict molecular motions to suppress the nonradiative decay of excited states, but also prevent the triplet exciton quenching by surrounding species (such as oxygen and moisture). A series of IRRTP systems was fabricated by doping a PVA matrix with different organic phosphors (Fig. 1c), followed by drop-casting the aqueous suspensions of these hybrid materials onto glass substrates and heating them at 65 °C for 5 h. Detailed experiments (Fig. 2b, Supplementary Methods, and Supplementary Figs. 1 and 2) indicate that the phosphorescence emission of SDP-doped PVA (0.3 mg/mL doping concentration) was significantly enhanced upon irradiation for 45 min, exhibiting the highest increase in phosphorescence lifetime from 58.03 to 828.81 ms (14.3 folds) after the irradiation. Each system showed

extraordinary phosphorescent emission after the irradiation (Fig. 2a, Supplementary Movie 1, Supplementary Figs. 3–5, and Supplementary Table 1) except the ABP-doped system (discussed later). Specifically, SDP, ODP, TDP, DP, 4,4-DB and BFPE-doped systems presented weak phosphorescence emission before the irradiation, while revealing green phosphorescence for 3–8 s after 45 min irradiation. 2,2-DB-doped PVA showed unique yellow phosphorescence emission up to 3 s after the irradiation.

Normally, strong intermolecular hydrogen bonding interactions could suppress the nonradiative transition rate of phosphors to some extent. However, weak phosphorescence emission was observed in this series of IRRTP systems before the UV irradiation, indicating that the IRRTP emission was not dominated by hydrogen bonding interactions. Combined with our previous work[34], the covalent bond (C–O–C) formation after the UV irradiation may be more reasonable to explain this irradiation-responsive phosphorescence emission. $^1$H NMR and Fourier-transform infrared spectra confirm the formation of strong intermolecular hydrogen bonding interactions in these systems (Supplementary Figs. 6 and 7). Upon increasing the irradiation time, a broadened peak located at 3264 cm$^{-1}$ and a slightly increased peak around 1141 cm$^{-1}$ prove the formation of the new ether bond (C–O–C). Deconvoluted C1$s$ peak at around 286.3 eV in the X-ray photoelectron spectroscopy (XPS) also supports the covalent C–O–C bond formation (Supplementary Fig. 8). Differential scanning calorimetry and thermogravimetric analysis provide additional evidence for the formation of a more rigid environment after UV irradiation (Supplementary Figs. 9 and 10).

To gain more insights into the mechanism of the irradiation-responsive phosphorescence, the photophysical properties of these hybrid films were further studied. Steady-state (prompt) photoluminescence spectra of the films (Supplementary Fig. 11) exhibit typical dual emission bands ranging from 300 to 500 nm. Time-resolved (delayed) photoluminescence spectra (Supplementary Fig. 12) of these films show increased phosphorescence intensity after the irradiation, directly revealing the character of irradiation-enhanced phosphorescence emission in these polymeric systems. Because of similar molecular structures of phosphors, the phosphorescence emission peaks of the irradiation-responsive polymeric films are in a similar wavelength range, reaching approximately from 480 to 550 nm. These films show no obvious change in the fluorescence lifetime, which is at the nanosecond level (Supplementary Table 2). In addition, the phosphorescence lifetime of phosphor powders, as well as doped films before and after 45 min irradiation was recorded (Supplementary Figs. 13 and 14). In the powder state, the phosphors show phosphorescence lifetimes on a millisecond scale (Fig. 2c). SDP, ODP, TDP, ABP, DP, 4,4-DB, 2,2-DP, and BFPE-doped films before prolonged irradiation exhibit phosphorescence lifetimes of 58.03, 38.54, 19.70, 26.59, 350.61, 21.59, 168.87, and 22.13 ms, which obviously increase to 828.81, 149.66, 70.70, 25.06, 480.18, 186.55, 257.06, and 222.09 ms after 45 min irradiation, respectively. Meanwhile, these films present relatively low phosphorescent quantum yields of 2.06%, 0.21%, 1.13%, 1.56%, 4.84%, 1.75%, 2.07%, and 0.07% under 254 nm excitation before prolonged irradiation, which increase obviously to 4.96%, 1.03%, 1.55%, 4.38%, 8.67%, 1.59%, 7.35%, and 1.16% after 45 min irradiation. Commission Internationale de L'Eclairage (CIE) coordinates (Supplementary Fig. 15) calculated from their delay spectra mainly show green and yellow emission colors. After 45 min irradiation, SDP, ODP, TDP, ABP, DP, 4,4-DB, 2,2-DP, and BFPE-doped films at ambient conditions give the CIE coordinates of (0.173, 0.313), (0.245, 0.378), (0.243, 0.385), (0.252, 0.325), (0.248, 0.352), (0.253, 0.415), (0.367, 0.531), and (0.185, 0.191), respectively. Above results indicate that

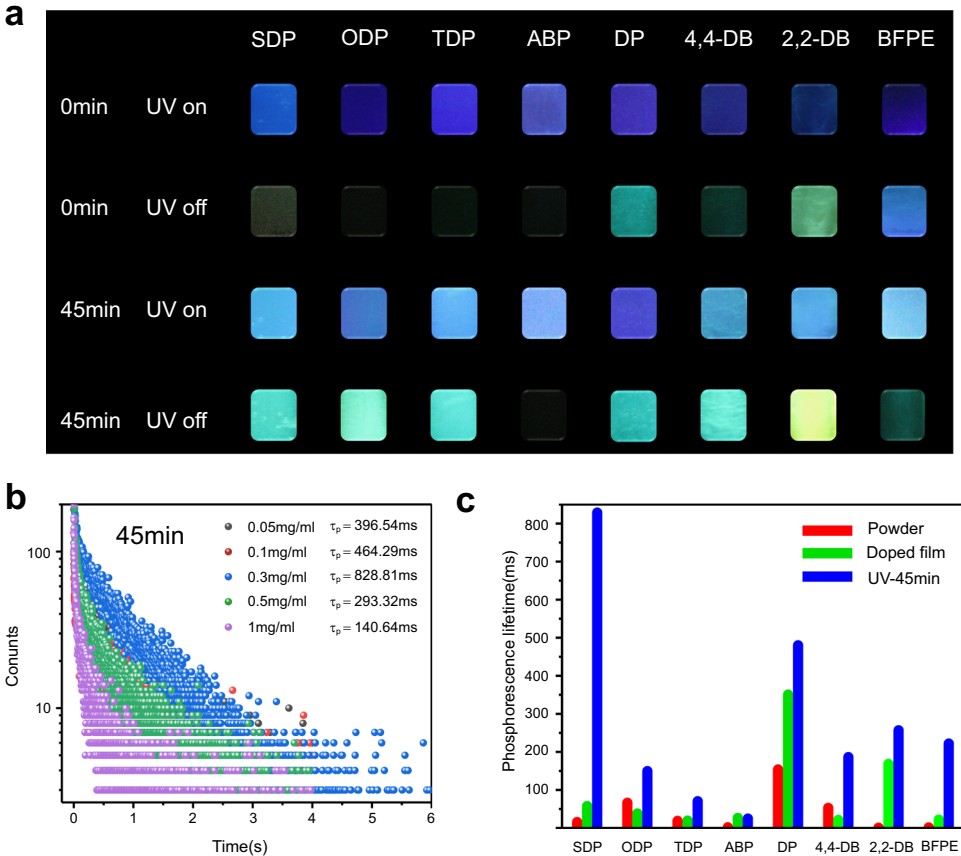

**Fig. 2 Irradiation-responsive room-temperature phosphorescence of the eight hybrid films. a** Photographs of the eight polymeric systems before and after turning off UV 254 nm light source. **b** IRRTP lifetime of the SDP-based polymeric system at different doping concentrations under 254 nm light irradiation for 45 min. **c** Comparison of room temperature phosphorescence lifetime of the eight polymeric systems at different states. Red bar: phosphor powder without the PVA matrix. Green bar: phosphor doped PVA films without the light irradiation. Blue bar: phosphor-doped PVA films with UV irradiation for 45 min.

irradiation not only enhances the phosphorescence lifetime, but also makes the color tunable.

The intriguing IRRTP is closely related to the rich –OH group in the PVA matrix as well as to the formation of covalent bonds under irradiation, providing a more rigid surrounding environment. To gain in-depth insight into the unique optical properties, a set of irradiation time-dependent experiments was conducted. Considering the conspicuous phosphorescence enhancement by irradiation, the SDP-based system was selected as a representative to shed light on the origin of the IRRTP. The two adjacent peaks at 405 and 460 nm in the fluorescence spectra (Supplementary Fig. 16) gradually shift to 362 and 481 nm during the irradiation time from 0 min to 5, 15, 30, 45, 60, 80, 100, and 120 min, respectively. Without the irradiation, the delay spectrum of the SDP film at ambient conditions has only one emission band at 408 nm. Upon increasing the irradiation time from 0 to 120 min, the emission intensity at 488 nm increases, while the emission intensity at 408 nm decreases in the delay spectrum (Fig. 3a), demonstrating good linearity on the CIE coordinate (Supplementary Fig. 15a). Meanwhile, SDP powder shows a phosphorescence lifetime of 15.91 ms, and SDP-doped film presents a prolonged emission of 58.03 ms. After 45 min irradiation, the SDP-doped film exhibits a 14.3-fold phosphorescence emission lifetime enhancement to 828.81 ms (Fig. 3b and Supplementary Figs. 16 and 17). In addition, three-dimensional scanning of excitation-phosphorescence emission on these films under ambient conditions was performed (Supplementary Figs. 18 and 19). For each film, the optimal excited and emission wavelengths

were in a similar area before and after the irradiation. In case of the SDP film (Fig. 3c), the main emission peak shows a significant bathochromic shift from 408 to 488 nm when the irradiation time increases from 0 min to 5, 45, and 120 min, which is accompanied by a phosphorescence emission color change from fugacious blue to extended green under ambient conditions (Fig. 3d).

To further verify the origin of the irradiation-responsive emission, we investigated the temperature effect on the delay spectra of the SDP-based film (Fig. 3e and Supplementary Fig. 20a). As the temperature decreases from 250 to 77 K, the main emission band lies at 394 nm before the irradiation. After 45 min irradiation, the emission band at 394 nm, as well as two new bands at 408 and 488 nm increase obviously (Supplementary Fig. 20b). The 2,2-DB-based film shows three major photoluminescence peaks at 409, 440, and 473 nm without the irradiation at 77 K, and three new peaks at 450, 506, and 537 nm appear after 45 min irradiation (Supplementary Figs. 21 and 22). To our surprise, the ABP-based film presents obvious irradiation-enhanced phosphorescence emission at 77 K (Supplementary Figs. 23 and 24d). Different emission bands before and after the irradiation may be caused by the simultaneous presence of multiple luminous centers. CIE coordinates further indicate the presence of various luminescent centers in these films (Supplementary Fig. 25). Because of a possible thermal dissipation caused by two methyl groups, the ABP-based film does not exhibit irradiation-responsive phosphorescence at room temperature.

As discussed above, the increase of the emission band intensity is due to the suppression of nonradiative transitions of the

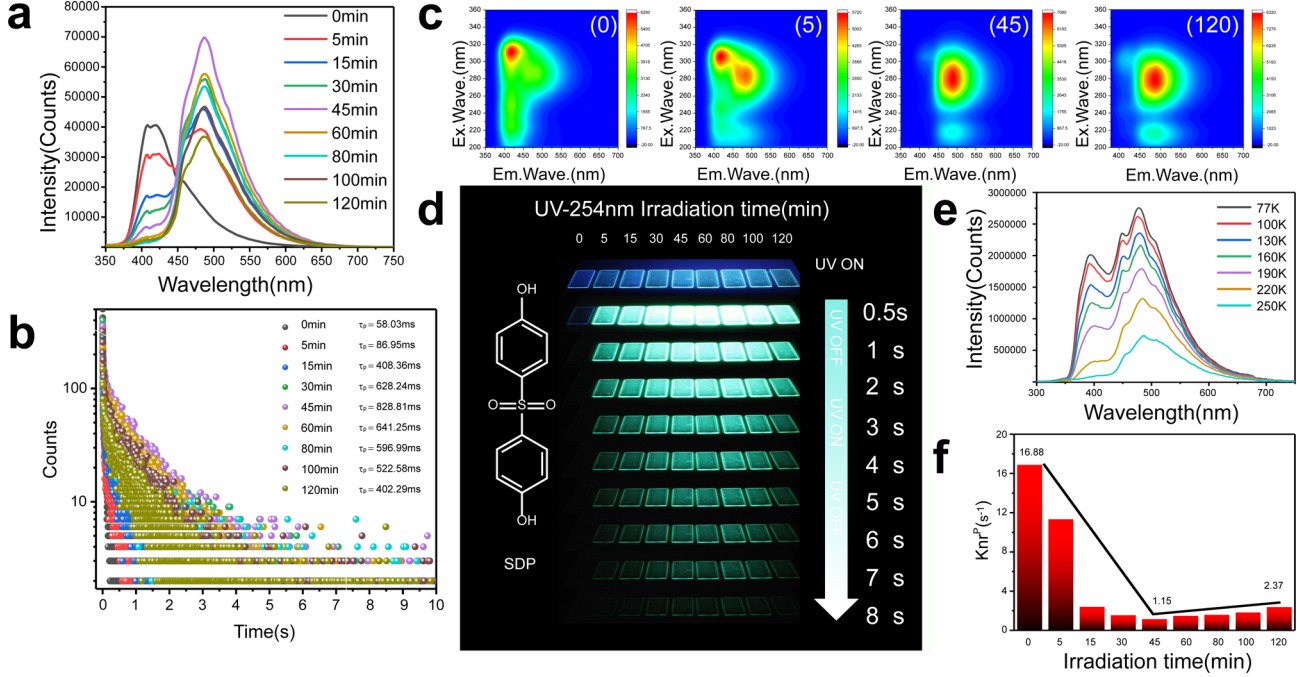

**Fig. 3 Photophysical properties of SDP-doped film. a** Time-resolved photoluminescence spectra and **b** phosphorescence decay curves under irradiation with 254 nm UV light for different irradiation times. **c** Excitation-phosphorescence emission mapping under 0, 5, 45, and 120 min continuous irradiation with 5 ms delay at room temperature. Em.Wave. emission wavelength, Ex.Wave. exitation wavelength. **d** Phosphorescence emission observed at different time intervals before and after switching off the light excitation at ambient conditions. **e** Phosphorescence spectra of SDP-doped film after 45 min irradiation measured at different temperatures from 77 to 250 K. **f** Nonradiative decay rate constant with different irradiation times.

excited state. Because of the spin-forbidden transition from the triplet excited state to the ground state, these molecules show very slow radiative ($k_r^P$) and nonradiative ($k_{nr}^P$) transition rates for green IRRTP emission, accompanied by fast fluorescence decay for strong and blue prompt emission[44]. For instance, $k_r^P$ of SDP with 45 min irradiation reaches $6.0 \times 10^{-2}\,\mathrm{s}^{-1}$ at room temperature (Supplementary Table 2), which is 6 fold lower than without the irradiation ($35.5 \times 10^{-2}\,\mathrm{s}^{-1}$). Meanwhile, the nonradiative rates ($k_{nr}^{phos}$) were calculated based on the lifetime and quantum yield of the SDP-based film with different irradiation times (Fig. 3f), indicating that $k_{nr}^{phos}$ plays a critical role in manipulating the phosphorescence lifetime. With 45 min irradiation, $k_{nr}^{phos}$ of the SDP-based film dramatically decreases to 1.15 s$^{-1}$, in sharp contrast with the value of 16.88 s$^{-1}$ at 0 min.

**Proposed mechanism for IRRTP.** Most of the pure organic molecules have singlet excitons with a very short lifetime. Only a small portion can realize the transition from an excited singlet state ($S_n$) to an excited triplet state ($T_n$), and then decay from the $T_1$ state to the $S_0$ state emitting phosphorescence with a short lifetime. Almost all of the prompt or delay emission bands of the eight films are located at the same wavelength (Supplementary Fig. 26). Based on above photophysical behavior before and after the irradiation, three main reasons causing the distinctive phenomena are proposed. Firstly, a large number of hydrogen bonding interactions which are formed in the doped systems suppresses intermolecular vibrations, so that the doped films show prolonged phosphorescence emission when compared to the powder state, as confirmed above (Figs. 2c, 4a, and Supplementary Fig. 6). Secondly, when hybrid films are irradiated by UV light, the hydroxyl groups in PVA are activated to form radicals capable of undergoing chemical reactions[45,46]. Oxygen radicals generated by the oxidation of hydroxyl groups can attack the PVA chain to form cross-linked ether structures. Thus, new covalent bonds (C–O–C) and complex cross-linking networks are

formed in the PVA matrix after the irradiation (Fig. 4b), further suppressing the nonradiative transitions and directly enhancing the phosphorescence in the systems. Upon increasing the irradiation time (Fig. 4d), two H protons in the benzene ring shift from 7.54 ppm (0 min) to 7.62 ppm (45 min) and from 6.70 ppm (0 min) to 6.77 ppm (45 min), respectively. Thirdly, because the hydroxyl groups in PVA chain and SDP are almost completely consumed by the formation of the cross-linking bonds (C–O–C) when the irradiation time exceeds 80 min, the previously formed hydrogen bonding is broken to release free hydroxyl groups, and thus these free hydroxyl groups continue to form new cross-linking bonds (Fig. 4c). The chemical shift of the two H protons in the NMR spectrum recover to 7.57 and 6.74 ppm when increasing irradiation time from 45 to 120 min, respectively. Although the formation of more cross-linking networks can further inhibit the thermal dissipation of the PVA chain, the reduction of the hydrogen bonding interactions between SDP and PVA would significantly increase the nonradiative transitions of the SDP molecule.

In order to exclude the influence of the oxygen consumption mechanism, several control experiments were conducted. Firstly, SDP-doped polymethyl methacrylate (PMMA) shows no obvious phosphorescence and irradiation-enhanced phosphorescence emission before and after 45 min irradiation, while obvious irradiation-enhanced phosphorescence emission occurs in SDP-doped PVA-87% (87% hydrolyzed PVA) and PVA-80% (80% hydrolyzed PVA) matrices (Supplementary Fig. 27). At the same time, SDP-doped poly(vinyl alcohol-co-ethylene) (PVA-co-PE) also exhibits obvious irradiation-enhanced phosphorescence emission, since PVA-co-PE is known to act as an oxygen barrier[47,48]. Meanwhile, SDP doped with polyvinyl acetate (PVAc) does not show irradiation-enhanced phosphorescence, because the absence of hydroxyl groups in the PVAc polymer prevents the formation of C–O–C covalent bonds under UV irradiation. Secondly, SDP-doped PMMA film presents very weak

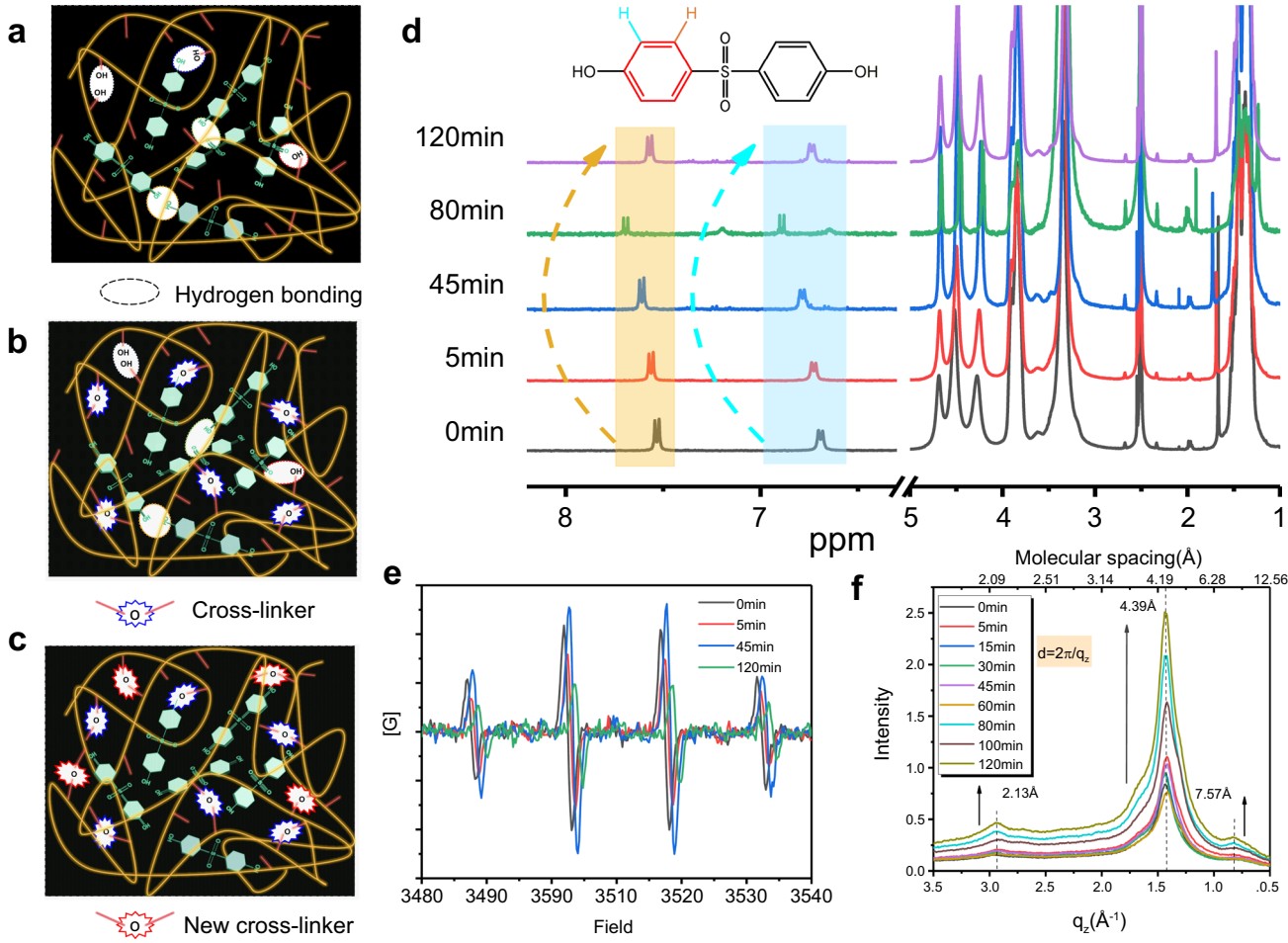

**Fig. 4 Mechanism of IRRTP at room temperature. a** Hydrogen bonding interactions between SDP and SDP, SDP and PVA, and PVA and PVA in unirradiated SDP-based film. **b** Covalent bond (C–O–C) formation between PVA chains after 254 nm UV light irradiation for <45 min. **c** Rearrangement of hydrogen bonding interactions, and further formation of covalent bond (C–O–C) after irradiation for more than 45 min. **d** $^1$H NMR spectra of SDP-based film (1 mg/mL) in DMSO-$d_6$ recorded after different irradiation times. **e** Electron paramagnetic resonance spectra and **f** one-dimentional scattering profiles in the $q_z$ direction of GiWAXS pattern for SDP-based film (0.3 mg/mL) recorded after different irradiation times.

photoluminescence and delayed phosphorescence emission intensity (Supplementary Figs. 28 and 29).

To obtain more insights into the mechanism, UV irradiation experiments were conducted in a glove box in which the oxygen level was maintained below 0.01 ppm. Obvious emission enhancement was observed after 45 min continuous UV-irradiation on SDP-doped PVA film, and the phosphorescence emission of irradiated and unirradiated films was distinguishable (Supplementary Fig. 29). In addition, an ethylene–vinyl alcohol copolymer-sealed oxygen barrier layer was deposited on top of the sample to avoid the exposure of the emitting layer to surrounding oxygen (sealed in the glove box to exclude oxygen entrapping). Phosphorescence emission photographs were recorded in the cuvette that was filled with oxygen (Supplementary Fig. 29b, c). Before the irradiation, SDP-doped PVA film and SDP-doped PVA film sealed with oxygen barrier show no visible phosphorescence. After 45 min irradiation, both films exhibit obvious green afterglow emission with a duration of ~9 s. These results indicate that the luminescence switching behavior of SDP-doped PVA is not closely related to the presence of oxygen.

As discussed above, ABP in the film state did not show obvious IRRTP after 45 min irradiation (2.13 ms at the power state, 26.59 ms after doped into a PVA matrix without the irradiation, and 25.06 ms after 45 min irradiation, Supplementary Figs. 13d and

14c). Unexpectedly, the phosphorescence lifetime increases to 2031.07 ms after 45 min irradiation, which is 1.5 folds higher than the case without the irradiation (1342.57 ms) at 77 K (Supplementary Fig. 24d). Even though two hydroxyl groups from ABP molecule can form hydrogen bonding interactions with PVA chains, since the vibration from methyl groups in ABP is still relatively strong, the irradiated system could not fully suppress nonradiative transition of ABP, and thus did not show irradiation-enhanced phosphorescence. Taken these results together, it can be concluded that the significant enhancement in the phosphorescence lifetime of these films should be ascribed to the suppression of nonradiative transitions by strong covalent bond formations.

The cross-linking microstructures formed in the films upon the irradiation time were further supported by scanning electron microscope (SEM) studies (Supplementary Figs. 30 and 31). Electron paramagnetic resonance (EPR) spectra were recorded to characterize the SDP-doped film before and after UV irritation at a wavelength of 254 nm (Fig. 4e). The film was subjected to the EPR experiments in the open air at room temperature, showing four peaks at 3486, 3501, 3516, and 3531 Gs. The noticeable EPR signals confirm that the film contains stable hydroxyl radical species before or after the irradiation[49]. Compared with the SDP-doped film before UV irradiation, the intensity increases after 45 min UV irradiation. These

phenomena suggest that the phosphor in PVA matrix has a radical character before UV irritation and the radical amount increases after UV irritation. Two-dimensional Grazing-incidence wide-angle X-ray scattering (GiWAXS) patterns (Supplementary Fig. 32) and one-dimensional scattering profiles in the $q_z$ direction (Fig. 4f and Supplementary Fig. 33) indicate multiple aggregated interactions in dry films[26]. Two new types of peaks at 2.13 and 7.57 Å were observed upon increasing the irradiation time, proving that there are also π–π stacking interactions in the cross-linked structures. Meanwhile, the scattering band at around 4.39 Å from the GiWAXS pattern of ABP-doped film indicates no obvious π–π stacking interaction formed upon increasing the irradiation time (Supplementary Fig. 34). This is a main reason that ABP-doped film has no irradiation-enhanced phosphorescence emission phenomenon.

According to the powder X-ray diffraction results (Supplementary Fig. 35), only one low-intensity broad diffraction band at around 19.41° was observed, which indicates the amorphous nature of these films. The position of this diffraction band can be varied upon the irradiation time, attributed to the cross-linked microstructures. The comparison of the UV–vis absorption spectra before and after the irradiation (Supplementary Fig. 36) shows that the π–π*-related absorption peak at 238 nm in SDP-doped and DP-doped films decreases gradually over 120 min irradiation, probably caused by the destruction of the stacked phosphor network in the matrix during the irradiation. Based on these results, we conclude that the irradiation-enhanced phosphorescence in this work not only depends on the molecular structures, but also hydrogen bonding, covalent bond (C–O–C) formation between PVA chains and phosphors, and π–π stacking interactions.

The steady-state photoluminescence spectra at 77 K (Supplementary Fig. 37) were also measured. Before the irradiation, the prompt emission band at 330 nm with an energy gap of 0.63 eV ($S_1$, 3.76 eV; $T_1$, 3.13 eV) between the $S_1$ and $T_1$ excited state levels ($\Delta E_{ST}$) can be obtained. After 45 min irradiation, the prompt emission band at 348 nm with the $\Delta E_{ST}$ value of 0.43 eV ($S_1$, 3.56 eV; $T_1$, 3.13 eV) was recorded. Simultaneously, the theoretical calculation of SDP-based film was conducted to verify the irradiation-enhanced phosphorescence (Supplementary Fig. 38), and the calculated $\Delta E_{ST}$ values of SDP-based film before and after the irradiation were 1.63 and 1.59 eV, respectively. Taken the experimental and theoretical results together, it was concluded that an appropriate energy gap from $S_1$ to $T_1$ state may make the ISC process efficient, thus facilitating the generation of IRRTP emission.

**Application studies of IRRTP systems.** Then, a series of flexible and transparent polymeric IRRTP films was successfully fabricated (Supplementary Fig. 39), and an inscribed sample showed a clear image in delayed emission after the excitation (group 1, element 6 in USAF test target, Supplementary Fig. 40)[48]. The absolute luminance of these films, excited under UV light (280 nm) at room temperature, remains distinguishable because of the afterglow which can be seen by naked eyes (Supplementary Table 3). Based on the IRRTP feature of the developed amorphous materials, we took one step further to demonstrate their application potential for polychromatic screen printing. As demonstrated in Fig. 5a, b, different patterns including pandas and lotuses were fabricated through a straightforward screen-printing technique by using eight phosphor-doped PVA systems. As a representative, the irradiation-enhanced photoluminescence of SDP-based and 2,2-DB-based films exhibits obvious green and yellow phosphorescence emission, respectively (Supplementary Movies 2 and 3). With exception of the ABP-based film which did not show the characteristics of

irradiation-enhanced phosphorescence at room temperature, lotus patterns with 15 different emission colors (fluorescence and phosphorescence) can be obtained.

Meanwhile, an application in multilevel information encryption was also explored (Fig. 5c). The information numbers "12345678" were patterned by using the eight phosphor-doped PVA systems as encryption inks, respectively, and the letters were dried at 65 °C in an oven for 30 min. Surprisingly, the phosphorescence emission color and brightness of these patterned letters change simultaneously upon increasing the irradiation time on the paper substrate. Without the irradiation, eight letters exhibit faint or invisible phosphorescence emission, and removing the excitation for 0.5 s leads to weak appearance of "12356". Then, when increasing the irradiation time for 15 and 45 min, the coded information gradually shows "1235678" and "12345678", respectively. Secondly, the changeable phosphorescence emission color provides another higher level of information security encryption. Different combination of green, blue, and yellow colored information emerges when increasing the irradiation time from 0 to 45 min. More specifically, without the irradiation, letters "13" and "256" show green and blue emission, respectively. After 15 min irradiation, green, blue, and yellow emission was observed from "138", "256", and "7", respectively. At 45 min irradiation, letters "12348", "5", and "67" present green, blue, and yellow emission, respectively. Significantly, ABP-doped PVA ink on the paper substrate shows weak but visible green phosphorescence emission after 45 min irradiation at room temperature. This observation may be a result of different compositions in each paper substrate (Supplementary Fig. 41). Such flexible, highly efficient, and irradiation-dependent IRRTP makes the developed systems ready for diverse applications, such as polychromatic screen printing, information storage, and multilevel anticounterfeiting without complicated fabrication processes.

## Discussion

In conclusion, we have developed a series of UV irradiation-responsive IRRTP systems through an efficient strategy of doping the selected phosphors within the PVA matrix. After continuous irradiation for 45 min, eight kinds of doped polymer films show irradiation-enhanced green or yellow phosphorescence emission. In particular, when increasing the irradiation time from 0 to 45 min, SDP-doped film presents striking green phosphorescence emission, showing 14.3-fold phosphorescence lifetime enhancement from 58.03 to 828.81 ms, and 2.4-fold phosphorescence quantum yield enhancement from 2.06% to 4.96%. Detailed experimental and theoretical calculation results demonstrate that there is significant suppression of the non-radiative transitions in these systems. Achieving the conditions necessary for generating long-lived phosphorescence at room temperature is the result of hydrogen bonding interactions and cross-linked bond formation. This study not only provides a design strategy to the preparation of irradiation-responsive phosphorescent systems, but also offers a guideline for developing advanced phosphorescent materials toward polychromatic screen printing, multicolor displays, multilevel information encryption, and biological applications.

## Methods

**Preparation of doped matrix.** PVA solid (6 g) was dissolved in deionized water (200 mL) at 95 °C for one hour, and subsequently filtered to obtain PVA aqueous solution (30 mg/mL) for further use.

**Preparation of hybrid films and patterning.** Firstly, SDP, ODP, TDP, ABP, DP, 4,4-DB, 2,2-DP, and BFPE (0.3 mg/mL for each phosphor) were, respectively, dispersed in eight vials containing PVA solutions with the concentration of

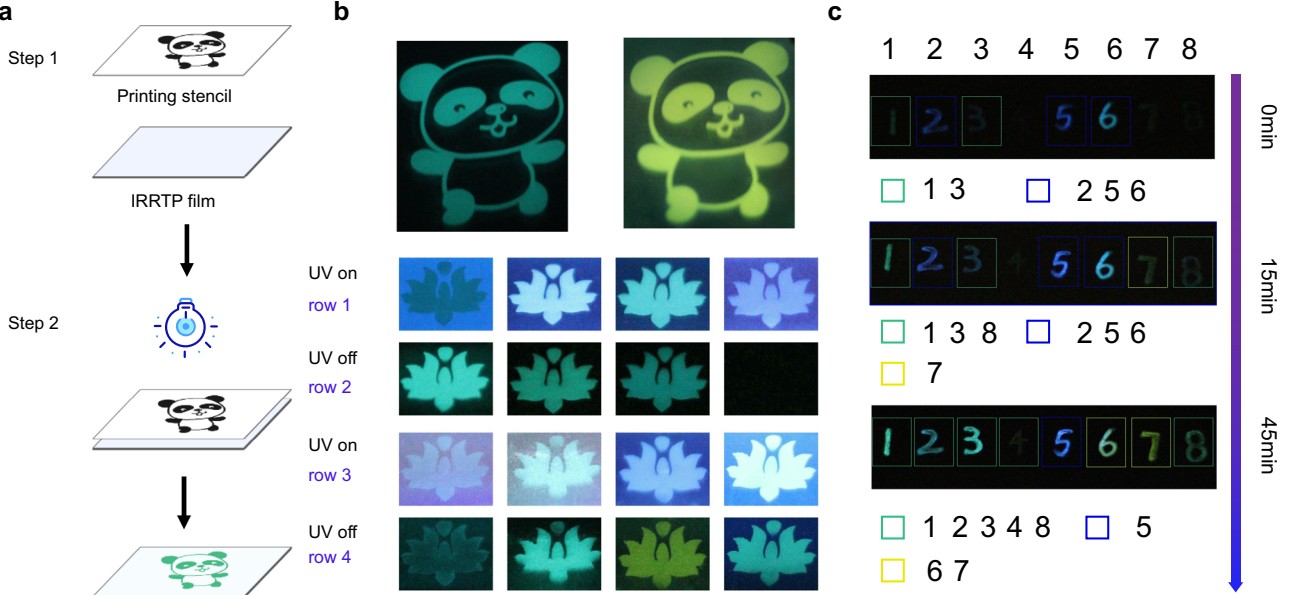

**Fig. 5 UV irradiation-responsive IRRTP for green screen printing and multilevel information encryption. a** Screen printing process for patterning. Flat IRRTP films were fabricated by drop-coasting the premixed solution on 75 mm × 25 mm glass substrate, followed by drying at 65 °C for 3 h. Then, the designed images were superimposed and fixed on the film surface. The obtained layers were continuously irradiated by a portable 254 nm UV lamp for 45 min to finish the printing progress. **b** IRRTP photographs of panda and lotus patterns. Green and yellow panda patterns were printed with SDP-based and 2,2-DB-based films, respectively. In row 1, from left to right, printed lotus by SDP, ODP, TDP, and ABP-based films under 254 nm UV lamp. In row 3, from left to right, printed lotus by DP, 4,4-DB, 2,2-DP, and BFPE-based films under 254 nm UV lamp. In rows 2 and 4, from left to right, corresponding printed lotus after turning off 254 nm UV lamp for 0.5 s. **c** Irradiation time-dependent anticounterfeiting photographs (after turning off UV light for 0.5 s) of eight doped films as the inks under ambient conditions. Numbers 1–8 were encrypted with SDP, ODP, TDP, ABP, DP, 4,4-DB, 2,2-DP, and BFPE-based films, respectively. Changeable encrypted information was shown by irradiation for 0, 15, and 45 min.

30 mg/mL. Eight homogeneous solutions were obtained after ultrasonication for 5 h, which were then left standing for one hour. Secondly, a series of films was fabricated by a drop-casting the prepared aqueous solutions on 75 mm × 25 mm glass substrates. Then, these films were dried at 65 °C for 3 h. Lastly, the designed images were superimposed and fixed on the film surface, and the double layers were irradiated by a portable UV-254 nm lamp for 45 min to generate the printing patterns.

## Data availability

All the other data supporting the findings of this study are available within the article and its supplementary information files and from the corresponding author upon reasonable request.

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

## Acknowledgements

This work was financially supported by the National Natural Science Foundation of China (21875025), the Special Program of Chongqing Science and Technology Commission (cstc2018jcyjAX0296 and cstc2017zdcyzdyfX0007), Innovation Research Group at Institutions of Higher Education in Chongqing (CXQT19027), the Science and Technology Research Program of Chongqing Municipal Education Commission (KJZD-K201801101), and the Postgraduate Tutor Team Project of Polymer Materials Engineering of Chongqing Education Commission. The research was also supported by the Singapore Agency for Science, Technology and Research (A*STAR) AME IRG grant (A1883c0005).

## Author contributions

Y.F.Z., L.G., X.Z., Z.H.W., and C.L.Y. conceived and were responsible for the experiments. Y.F.Z., C.L.Y., and Y.L.Z. wrote the manuscript. H.L.T., L.J.Q., and Y.B.L. conducted the microscopic experiments and analysis. C.L.Y. contributed to the theoretical calculations.

## Competing interests

The authors declare no competing interests.
