## [Peer Review File · Nature Communications]

REVIEWER COMMENTS

Reviewer #1 (Remarks to the Author):

In this work, the authors reported an ultraviolet irradiation-responsive dynamic ultralong organic phosphorescent system based on irradiation induced polymerization. With such new strategy, the nonradiative transitions and triplet excitons quenching rate within oxygen and moisture, can be efficiently suppressed. As the result, the film RTP lifetime can be increased by 14 folds, which is really meaningful for gaining ultralong lifetime RTP materials.

In addition, the authors have carefully summarized the development of the RTP research area and the main problems in this area.

Therefore, I recommend this manuscript to be published.

Reviewer #2 (Remarks to the Author):

This manuscript presents a feasible and facile strategy to achieve ultraviolet irradiation-responsive ultralong room-temperature phosphorescence (IR RTP) in several simple amorphous polymer materials. Extraordinary IR RTP in the amorphous state endows them with longer phosphorescence lifetime and higher phosphorescence quantum yield. It is an interesting work with novel ideas and commendable strategies. However, there are still some revisions should be made in this manuscript:

- 1) The mechanism description of IR RTP should be complemented in the manuscript. Besides, the extraordinary achievements in phosphorescence lifetime and phosphorescence quantum yield was not explained conclusively which need further statements.
- 2) What was the relationship among the eight compounds presented in the manuscript? What differences will be made by the differed structures of the compounds? Will the similar phenomenon of IR RTP occur in other compounds with resembled structures? The authors should make further explanations to summary the functions of compounds in the manuscript.
- 3) How to explain the further shift of H protons from 45 min to 80 min in Figure 4d?
- 4) Please recheck the manuscript and keep the writing in the same verb tense.
- 5) The author should provide reproducible experimental data in the main text to avoid the influence of impurities and the randomness of photo reaction.
- 6) In Fig.2a, comparing to original state, the afterglow of eight compounds red/blue shifted after the UV irradiation. The author needs to explain the specific reasons for the different phenomena.
- 7) In page 7, line 137. The author assumes that there is a covalent bond (C-O-C) formation after the UV irradiation, which requires a model reaction of eight compounds and vinyl alcohol/ethanol to prove the author's point of view.
- 8) The author can try other hydroxyl-containing polymers except PVA to verify whether there are similar phenomena.
- 9) In Fig. 4d, after 45 minutes of irradiation, the chemical shift of two H atoms in the benzene ring is 0.07 ppm, which is too small to be vigorously provable.
- 10) In Fig. 4d, since the formation of "C-O-C" covalent bond is irreversible, the author may need to explain what caused the two H atoms in the benzene ring recovering to the shift exactly as them before irradiation, when irradiation time increased to 120 min.
- 11) In Fig. 2a, the phosphorescence lifetime of ABP doped film deceased from 26.59 ms to 25.06 ms, after 45 min irradiation, which showed very significant contrast compared with other doped

systems. Corresponding explanation is needed.

Reviewer #3 (Remarks to the Author):

In this paper, the authors reported the intriguing phenomenon of organic room temperature phosphorescent materials, i.e. light-induced phosphorescence enhancement. By various experiments, authors claimed that this phenomenon originates from the UV-induced covalent bond formation (e.g. C-O-C) of PVA matrix polymers which greatly reduces non-radiative decay processes of the phosphors.

While authors argued that this is the first report for light-induced phosphorescence enhancement, two reports for a similar phenomenon have already been published as follows, which greatly reduce the novelty of this work:

- 1) S. Reineke and coworkers, *Sci. Adv.* 7310 (2019)
- 2) J. Kim and coworkers, *Adv. Opt. Mater.* 2000654 (2020)

In those papers, authors claimed that the origin of the phenomenon stems from the conversion of triplet oxygen to singlet oxygen by UV-irradiation, leading to the unique phosphorescence enhancement of organic phosphors, which definitely challenges arguments in the current manuscript.

This reviewer believes that the experiments conducted in the current work would not fully rule out the possibility of the hypothesis suggested by previous works (i.e. UV-induced oxygen consumption and hence phosphorescence enhancement in the polymer matrix).

Considering those factors, this reviewer cannot support this work to be published in a premier journal such as *Nat. Commun.*

Response to Reviewer #1's Comments:

In this work, the authors reported an ultraviolet irradiation-responsive dynamic ultralong organic phosphorescent system based on irradiation induced polymerization. With such new strategy, the nonradiative transitions and triplet excitons quenching rate within oxygen and moisture, can be efficiently suppressed. As the result, the film RTP lifetime can be increased by 14 folds, which is really meaningful for gaining ultralong lifetime RTP materials.

In addition, the authors have carefully summarized the development of the RTP research area and the main problems in this area.

Therefore, I recommend this manuscript to be published.

Response: Thanks for your professional comments and kind recommendation of this work.

Response to Reviewer #2's Comments:

This manuscript presents a feasible and facile strategy to achieve ultraviolet irradiation-responsive ultralong room-temperature phosphorescence (IRRTTP) in several simple amorphous polymer materials. Extraordinary IRRTTP in the amorphous state endows them with longer phosphorescence lifetime and higher phosphorescence quantum yield. It is an interesting work with novel ideas and commendable strategies. However, there are still some revisions should be made in this manuscript:

Response: Thank you for your recognition of this work. According to the comments, we have carefully revised the manuscript. Please refer to the detailed response shown below.

Comment 1: The mechanism description of IRRTTP should be complemented in the manuscript. Besides, the extraordinary achievements in phosphorescence lifetime and phosphorescence quantum yield was not explained conclusively which need further statements.

Response: Thanks for the useful advice. In the "Proposed mechanism for IRRTTP" section of the previous version, the IRRTTP is proven to be resulted from the combined effects of hydrogen bonding and newly formed cross-linked covalent bond (C-O-C), and significant suppression of the nonradiative transitions in these hybrid film systems, preventing the quenching of triplet excitons by oxygen and moisture after long irradiation. At the same time, the phosphorescence lifetime and quantum yield were also achieved (page 9): *"SDP, ODP, TDP, ABP, DP, 4,4-DB, 2,2-DP, and BFPE doped films without the irradiation showed the phosphorescence lifetime of 58.03, 38.54, 19.70, 26.59, 21.52, 21.59, 168.87, and 22.13 ms, which obviously increased to 828.81, 149.66, 70.70, 25.06, 212.73, 186.55, 257.06, and 222.09 ms after 45 min irradiation, respectively. Meanwhile, these films exhibited relatively low phosphorescent quantum yields of 2.06, 0.21, 1.13, 1.56, 4.84, 1.75, 2.07, and 0.07 % under 254 nm excitation before the irradiation, and then increased obviously to 4.96, 1.03, 1.55, 4.38, 8.67, 1.59, 7.35, and 1.16 % after 45 min irradiation."* Because the phosphorescence quantum yield (Φ_p) and lifetime (τ_p) are calculated as $\Phi_p = \Phi_{ISC}K_p\tau_p = \Phi_{ISC}[1 - (K_{nr} + K_q)]\tau_p$ and $\tau_p = 1/(K_p + K_{nr} + K_q)$. Thus, phosphorescence quantum yield (Φ_p) and lifetime (τ_p) would increase when K_{nr} is largely suppressed by hydrogen bonding and cross-linked covalent bond.

In addition, we have revised the sentences in the "Conclusion" section: *"Especially, when increasing the irradiation time from 0 min to 45 min, SDP-doped films present striking green phosphorescence emission, increased phosphorescence lifetime from 58.03 ms to 828.81 ms with 14-folds of enhancement, and increased phosphorescence quantum yield from 2.06 % to 4.96 % with 2.4-folds of enhancement. Based on various experimental and*

theoretical calculation results, it has been demonstrated that there is significant suppression of the nonradiative transitions in these systems”

As suggested, we have further established the relationship among the eight compounds by adding the reasons for the shift of H proton signals in the benzene ring, the C-O-C covalent bond formation process, and oxygen effect on the IRRTP emission. The related discussions have been added to the revised manuscript.

For the relationship among the eight compounds (page 5 to page 6): *“For the eight pure organic phosphors containing hydroxyl group (O-H), sulfone group (O=S=O), or aromatic carbonyl group (Ar-C=O), these functional groups could form abundant hydrogen bonding or import a strong SOC effect to promote the ISC. Meanwhile, the nonradiative decay could efficiently suppress by multiple hydrogen bonding, and further suppressed by cross-linked covalent bond (C-O-C) after 45 min UV-254 nm irradiation. At the same time, the molecular structures of C-(O=S=O)-C, C-O-C, C-S-C, and C-C(CH₃)₂-C between edge benzene rings in SDP, ODP, TDP, and ABP show different degrees of steric effect, respectively. For example, the functional sulfone group (O=S=O) in SDP can import a strong SOC effect to promote the ISC, while larger group of C-C(CH₃)₂-C would lead to obvious steric effect in ABP.”*

For the C-O-C covalent bond formed process (page 14): *“Secondly, when hybrid films were irradiated by UV light, PVA hydroxyl groups are oxidized and dehydrated to form radicals. Oxygen radical generated by oxidation of hydroxyl group may attack the PVA main chain to form cross-linked ether structure.”*

For the reason of the H protons further shift in the benzene ring (page 14 to page 15): *“Thirdly, because of hydroxyl groups in PVA chain and in SDP are almost completely consumed by forming the cross-linked bonds (C-O-C) when the irradiation time exceeds 80 min, the former formed hydrogen bonding would be broken to release free hydroxyl groups, and subsequently, these free hydroxyl groups continue to form new cross-linked bonds.”*

Comment 2: What was the relationship among the eight compounds presented in the manuscript? What differences will be made by the differed structures of the compounds? Will the similar phenomenon of IRRTP occur in other compounds with resembled structures? The authors should make further explanations to summary the functions of compounds in the manuscript.

Response: Thanks for your professional suggestion. To achieve high performance ultralong room temperature phosphorescence, efficient intersystem crossing (ISC_(S₁-T₁)), efficient radiative transition from T₁ to S₀, and suppression or reduction of nonradiative decay and quenching of T₁ should be included. Combined with these strategies, after screening many kinds of potential phosphors in our lab, we fortunately found the ultralong RTP property of SDP-based phosphor doped-PVA matrix with obvious UV irradiation-enhanced effect. Thus, a series of resemble phosphors with irradiation-enhanced phosphorescence was also explored (ODP, TDP, ABP, DP, 4,4-DB, 2,2-DB, and BFPE).

Based on our previous work (Sci. Adv. 2018, 4, eaas9732), it was concluded that the ultralong phosphorescence emission after UV irradiation is not only resulted from hydrogen bonding, but also caused by C-O-C bond formation. In the "Introduction" section, for the eight pure organic phosphors containing hydroxyl group (O-H), sulfone group (O=S=O), or aromatic carbonyl group (Ar-C=O), these functional groups could form abundant hydrogen bonding or import a strong SOC effect to promote the ISC. Meanwhile, the nonradiative decay could be suppressed by multiple hydrogen bonding and further suppressed by covalent bond (C-O-C) after 45 min UV-254 nm irradiation. As shown in Figure R1, firstly, the molecular structures of C-(O=S=O)-C, C-O-C, C-S-C, and C-C(CH₃)₂-C between edge benzene rings in SDP, ODP, TDP, and ABP show different degrees of steric effect, respectively. For example, the functional sulfone group (O=S=O) in SDP can import a strong SOC effect to promote the ISC. Thus, the phosphorescence emission lifetime of SDP is longer than that of ODP and TDP. DP shows obvious phosphorescence emission before and after irradiation, and 4,4-DB, 2,2-DB, and BFPE doped systems exhibit shorter phosphorescence emission than SDP. These results further indicate that hydrogen bonding and cross-linked covalent bond C-O-C between PVA chain and phosphors play an important role for irradiation-enhanced phosphorescence. In a word, irradiation-enhanced phosphorescence in this work is closely relative to molecular structures of phosphors and polymer matrix.

Secondly, in order to gain more insights into the intriguing phenomenon of organic room temperature phosphorescent materials, two-dimensional Grazing-incidence wide-angle X-ray scattering (GiWAXS) patterns (Figure R2, i.e., Supplementary Fig. 32) and one-dimensional scattering profiles in the q_z direction (Figure R3, i.e., Supplementary Fig. 33) of SDP films were conducted. Two new types of peaks at 2.13 Å and 7.57 Å were observed upon increasing the irradiation time, proving that there are also π - π stacking interactions in the cross-linked structures. However, in Figure R4 (Supplementary Fig. 34), the scattering band at around 4.39 Å from two-dimensional GiWAXS of ABP film indicates no obvious π - π stacking interactions upon increasing the irradiation time. This is a main reason that ABP film has no irradiation-enhanced phosphorescence emission property.

Lastly, there is a fact that phosphorescence emission is affected by many factors, such as intermolecular processes including energy transfer, energy migration, and oxygen quenching, as well as the intramolecular nonradiative transition process. Thus, in Figure R5, no irradiation-enhanced phosphorescence was observed when we conducted other six compounds doped into PVA matrix. Combined with GiWAXS results of ABP films, we could conclude that *"the irradiation-enhanced phosphorescence in this work not only depends on the molecular structure, but also hydrogen bonding, cross-linked covalent bond C-O-C between PVA chain and phosphors, and π - π stacking interactions."* The related discussions have been added into the revised manuscript (page 18).

Figure R1. Eight phosphors from left to right (SDP, ODP, TDP, ABP, DP, 4,4-DB, 2,2-DB, and BFPE). (a) Molecular structures and (b) photographs of the eight doped polymeric systems before and after turning off UV 254 nm light source.

Figure R2 (Supplementary Fig. 32). 2D GiWAXS out-of-plane patterns of SDP doped film at 0.3 mg / mL doping concentration after irradiation for (a) 0 min, (b) 5 min, (c) 45 min, and (d) 120 min.

Figure R3 (Supplementary Fig. 33). 1D scattering profiles in the q_z direction of SDP doped film at 0.3 mg/mL doping concentration after irradiation from 0 min to 120 min.

Figure R4 (Supplementary Fig. 34). 2D GiWAXS out-of-plane patterns of ABP doped film at 0.3 mg/mL doping concentration after irradiation for (a) 0 min and (b) 45 min. (c) 1D scattering profiles in the q_z direction of ABP doped film for 0 min and 45 min.

Figure R5. Six resembling compounds (Compound 1, Compound 2, Compound 3, Compound 4, Compound 5, and Compound 6). (a) Molecular structures and (b) photographs of the six

doped polymeric systems with 0 min and 45 min irradiation before and after turning off UV 254 nm lamp.

Comment 3: How to explain the further shift of H protons from 45 min to 80 min in Figure 4d?

Response: We thank the reviewer for raising this important question. As we known, phosphorescence emission is determined by many factors, such as chemical structure, intra-/inter-molecular interactions, surrounding environment, et al. In this manuscript, for example, SDP-doped film has the longest phosphorescence emission lifetime after UV-254 nm irradiation for 45 min. However, phosphorescence emission lifetime of SDP-doped film becomes shorter after irradiation up to 80 min and 120 min than that of irradiation for 45 min.

Figure R6. ^1H NMR spectra of SDP-based film (**a.** 1 mg/mL, **b.** 3 mg/mL) in DMSO-d_6 under different irradiation times.

There have two main reasons for that. Firstly, a lot of hydroxyl groups in the form of hydrogen bonding exist in the film even after 45 min irradiation. Free hydroxyl group would decrease with more C-O-C covalent bond formation by longer irradiation time, and thus the H protons would further shift with increasing irradiation time up to 80 min. However, with the irradiation time increasing to 120 min, the former formed hydrogen bonding would be broken to release free hydroxyl groups, and subsequently, these free hydroxyl groups continue to form new cross-linked bonds. Secondly, from one-dimensional scattering profiles in the q_z direction of GiWAXS pattern (Figure R2c, i.e., Supplementary Fig. 32c), it is easy to find multiple aggregation interactions in films. The two new types of peaks at 2.13 Å and 7.57 Å were observed upon increasing the irradiation time, proving that there are also π - π stacking interactions formed in the cross-linked structures. The shift of H protons could also be affected by shielding effect because of more intra-/inter-molecular π - π stacking interactions formed. Lastly, to avoid any random results, we remeasured the ^1H NMR spectra of SDP-doped films with 0 min to 120 min irradiation (Figure R6). Same to the previous results, the H protons in the benzene ring keep shifted to downfield from 0 min to 80 min irradiation.

Comment 4: Please recheck the manuscript and keep the writing in the same verb tense.

Response: We appreciate the reviewer's constructive suggestion. These different verb tenses have been corrected. The manuscript has also been reviewed carefully in an effort to minimize and hopefully eliminate all other language errors. In addition, the Y-axis and Figure legends have been corrected in Figure 2b, Figure 4f, and Figure S12e.

Comment 5: The author should provide reproducible experimental data in the main text to avoid the influence of impurities and the randomness of photo reaction.

Response: Many thanks, this is a very important suggestion. To prove that the irradiation-dependent phosphorescence is not caused by the influence of impurities and the randomness of photoreaction, we have purified the organic phosphors after several recrystallization processes. SDP, ODP, TDP, ABP, DP, 4,4-DB, 2,2-DB, and BFPE were recrystallized by water, water, water, toluene, ethanol, ethanol, and ethanol, respectively. Here, we take SDP as an example to give further statement. Firstly, direct comparison of the phosphorescence emission for eight recrystallized phosphor-doped films at 0.3 mg/mL doping concentration is shown in Figure R1. From the photographs of the purified (Figure R1) and unpurified (Figure 2a in manuscript) doped systems before and after turning off UV 254 nm light source, both purified and unpurified doped systems all show similar intriguing phosphorescence. Secondly, the delayed photoluminescence spectra and phosphorescence decay curves of SDP-doped film were studied. In Figure R7c,d, the emission peaks (408 nm and 488 nm) of SDP-doped film before and after irradiation are at the same positions. In addition, the phosphorescence lifetime before and after UV 254 nm irradiation shows no major changes. Meanwhile, ODP-doped films have similar phosphorescence emission spectra before and after irradiation, and the phosphorescence lifetime of ODP-doped film before and after irradiation does not show major changes. Thirdly, in Figure R8, under the mobile phase of methanol/water, impurity peaks from the unpurified and recrystallized SDP and ABP were not detected by HPLC using the XB-C18 column.

Taken together, these results prove that the intriguing irradiation-dependent phosphorescence in this work does not originate from the influence of impurities and the randomness of photoreaction.

Figure R7. Photophysical properties of SDP-doped PVA (100% hydrolyzed) at 0.3 mg/mL doping concentration with 0 min and 45 min irradiation before and after recrystallization. (a) SDP at powder state. (b) SDP at recrystallized state. (c,e) Delayed photoluminescence emission spectra of SDP and ODP under different conditions. (d,f) Phosphorescence decay curves of SDP and ODP under different conditions.

Figure R8. HPLC curves of powder and crystals monitored at 254nm. Left: SDP; Right: ABP.

Comment 6: In Fig.2a, comparing to original state, the afterglow of eight compounds red/blue shifted after the UV irradiation. The author needs to explain the specific reasons for the different phenomena.

Response: Thanks for your constructive comment. From the Jablonski diagram, upon photoexcitation, a molecule in the ground state (S_0) is excited to a singlet state (S_n , $n \geq 1$). Subsequently, the singlet exciton transfers to the lowest singlet state (S_1) via fast internal conversion (IC) based on the Kasha rule. On one hand, red/blue shifts after the UV irradiation of SDP, DP and BFPE could be directly observed in Figure 2a, and ODP, TDP, 2,2-DB, 4,4-DB didn't show any color changes. Meanwhile, Figure R9 (Supplementary Fig. 12) and Figure R10 (Supplementary Fig. 26) also indicate that the phosphorescence emission peaks of eight compounds doped with PVA matrix have almost no changes, indicating that the irradiation-enhanced phosphorescence is resulted from the largely suppressed nonradiative transition in this work. On the other hand, we know that afterglow emission is tightly related with triplet state energy level. The energy level of excited triplet state (T_n) could be affected by the state of phosphors such as crystal state, solution state, isolated state, aggregated state, molecular conformation, etc. Upon increasing the irradiation time, SDP shows the most obvious red-shifted phosphorescence emission from blue to green. The main reason is the π - π stacking interactions of SDP molecule in PVA matrix. In Figure R2 (Supplementary Fig. 32), combined with GiWAXS results of SDP films, it proves that there are more π - π stacking interactions formed in the cross-linked structures. In Figure R4 (Supplementary Fig. 34), the scattering band at around 4.39 Å of two-dimensional GiWAXS of ABP film indicates that no other π - π stacking interactions can be detected upon increasing the irradiation time. In the ^1H NMR spectra of SDP film (Figure R6), when increasing the irradiation time, H protons in ^1H NMR spectra of ABP and 2,2-DB films have no obvious

changes (Figure R11), also proving that the red shift of SDP film as well as the same color change of 2,2-DB film after irradiation are related with the phosphor conformation and state. Thus, it can be concluded that the state of phosphors leads to different excited triplet state energy levels, which further affect the phosphorescence emission.

Figure 2a. Photographs of the eight polymeric systems before and after turning off UV 254 nm light source.

Figure R9 (Supplementary Fig. 12). Delayed photoluminescence spectra of (a) SDP, (b) ODP, (c) TDP, (d) ABP, (e) DP, (f) 4,4-DB, (g) 2,2-DB, and (h) BFPE doped films at 0.3 mg/mL doping concentration with 0 min and 45 min irradiation at room temperature.

Figure R10 (Supplementary Fig. 26). Normalized prompt and 5 ms delayed photoluminescence spectra of (a) SDP, (b) ODP, (c) TDP, (d) ABP, (e) DP, (f) 4,4-DB, (g) 2,2-DP, and (h) BFPE doped films at 0.3 mg/mL doping concentration at room temperature (dash line: without irradiation, solid line: with 45 min irradiation).

Figure R11. ^1H NMR spectra of (a) ABP-doped film and (b) 2,2-BC-doped film at 1 mg / mL in DMSO- d_6 under different irradiation times.

Comment 7: In page 7, line 137. The author assumes that there is a covalent bond (C-O-C) formation after the UV irradiation, which requires a model reaction of eight compounds and vinyl alcohol/ethanol to prove the author's point of view.

Response: Thanks for your professional and constructive comments. As we known, hydroxyl groups in PVA chain can be easily oxidized and dehydrated to form radicals under UV light, which may then cause other chemical reactions. Oxygen radical generated by oxidation of hydroxyls group would attack the PVA chain to form a cross-linked ether structure. This typical chemical reaction has been reported in some literature papers, such as *ACS Sustainable Chem. Eng.* 2016, 4, 2252-2258; *Polym. Degrad. Stab.* 2015, 114, 30-36; *J. Phys. Chem. B* 2017, 121, 1148-1157; *Fibers Polym.* 2014, 15, 101-107. The reaction mechanism of PVA when irradiated by UV is shown in **Figure R12**, where cross-linked ether structure can be formed between PVA chains, and between PVA and phenolic hydroxyl groups.

According to your suggestion, in order to prove this process, electron paramagnetic resonance (EPR) spectra of SDP-based film with different irradiation times were conducted in Figure R13. Pure PVA, SPD powder and SDP-doped PVA film showed no EPR signals. After UV irradiation, obvious radical character was detected by EPR. Meanwhile, ^1H NMR spectra of SDP-based film (15 mg/mL) in DMSO- d_6 with different irradiation times were conducted (Figure R14). The intensity of H proton from -OH group in SDP decreases upon increasing the irradiation time from 0 hour to 20 hours under UV-254 nm irradiation, indicating that the SDP molecule is gradually consumed due to the cross-linked bond (C-O-C) formation between PVA and SDP. Combined with these results, the formation of new cross-linked bonds (C-O-C) between PVA chain and SDP can be proven by radical reaction under the UV irradiation.

Figure R12. Reaction mechanism of PVA and SDP under UV irradiation.

Figure R13. Electron paramagnetic resonance spectra of SDP-based film (0.3 mg/mL) under different irradiation times without heating treatment. (a) without irradiation. (b) with 10 hours of irradiation by UV 254 nm.

Figure R14. ^1H NMR spectra of SDP-based film (15 mg/mL) in DMSO-d_6 under different irradiation times by UV 254 nm.

Comment 8: The author can try other hydroxyl-containing polymers except PVA to verify whether there are similar phenomena.

Response: According to your good suggestion, SDP phosphor was doped into six common polymer matrices including PVA-100%, PVA-87%, PVA-80%, PVA-co-PE, PVAc, and PMMA. As shown in **Figure R15** (Supplementary Fig. 27), these SDP doped hydroxyl-containing polymers exhibit a certain degree of irradiation-enhanced phosphorescence. Be more concrete, the SDP doped PVA 87%, PVA 80%, and PVA-co-PE films show no phosphorescence emission before the irradiation, while relatively obvious irradiation-enhanced phosphorescence after 45 min UV irradiation. Meanwhile, SDP doped PVA-100% exhibits higher phosphorescence emission than that of SDP doped PVA-87% and PVA-80%. As a reported oxygen-barrier, SDP doped PVA-co-PE matrix also shows obvious irradiation-enhanced phosphorescence after 45 min UV irradiation. This rules out that the enhanced-phosphorescence is caused by oxygen consumption in polymer matrix. Furthermore, SDP-doped PVAc and PMMA films have no phosphorescence emission before and after UV 254 nm irradiation. These results indicate that the hydrogen bonding and cross-linked bond (C-O-C) formations from hydroxyl groups in the hydroxyl group-rich polymers are the key reason for efficient irradiation-enhance phosphorescence.

Figure R15 (Supplementary Fig. 27). (a) Six common polymer matrices including 100% hydrolyzed PVA (PVA), 87% hydrolyzed PVA (PVA-87%), 80% hydrolyzed PVA (PVA-80%), poly(vinyl alcohol-co-ethylene) (PVA-co-PE), polyvinyl acetate (PVAc), and polymethyl methacrylate (PMMA). (b) Photographs of SDP-doped polymer matrices before and after turning off UV 254 nm light (left: without irradiation; right: with 45 min UV irradiation by UV 254nm).

Comment 9: In Fig. 4d, after 45 minutes of irradiation, the chemical shift of two H atoms in the benzene ring is 0.07 ppm, which is too small to be vigorously provable.

Response: Many thanks for your question. As observed in **Figure R16** (**Figure 4d**), two H protons in the benzene ring shift from 7.54 ppm (0 min) to 7.62 ppm (45 min) and from 6.70 ppm (0 min) to 6.77 ppm (45 min), respectively. The chemical shifts of the two H atoms in the benzene ring are 0.08 ppm and 0.07 ppm, respectively. To further confirm this, ^1H NMR spectra of SDP doped films with concentrations of 1 mg / mL and 3 mg / mL have been measured in **Figure R6**. For 1mg / mL concentration, two H protons in the benzene ring shift from 6.71 ppm (0 min) to 6.78 ppm (45 min) and from 7.63 ppm (0 min) to 7.66 ppm (45 min), respectively. The shifts are 0.06 ppm and 0.03 ppm, respectively. For 3 mg / mL concentration, two H protons in the benzene ring shift from 6.81 ppm (0 min) to 6.85 ppm (45 min) and from 7.55 ppm (0 min) to 7.61 ppm (45 min), respectively. The shifts are 0.04 ppm and 0.06 ppm, respectively. These results reveal that H proton shifts in the benzene ring do exist, dependent on the contraction. Although the value of chemical shift is relatively small, we suppose that the phenomenon is ascribed to the π - π stacking interactions. 2D GiWAXS out-of-plane patterns and 1D scattering profiles in the qz direction of SDP doped film in **Figure R2** and **R3** further confirm this point view. In addition, some reported results show similar small chemical shifts due to π - π stacking interactions (*Langmuir* 2010, 26,

16818-16827; *Angew. Chem. Int. Ed.* 2017, 56, 11252-11257; *J. Am. Chem. Soc.* 2014, 136, 5416-5423).

Combined with the above results, it is reasonable that the two H atoms in the benzene ring show relatively small chemical shifts after the irradiation by UV 254 nm.

Figure R16 (Fig. 4d) ^1H NMR spectra of SDP-based film (1 mg/mL) in DMSO-d_6 under different irradiation times.

Comment 10: In Fig. 4d, since the formation of “C-O-C” covalent bond is irreversible, the author may need to explain what caused the two H atoms in the benzene ring recovering to the shift exactly as them before irradiation, when irradiation time increased to 120 min.

Response: Many thanks for this valuable question. As you mentioned, the covalent bond C-O-C is irreversible. The chemical shift of H atoms in the benzene ring recovers back to the original position after 120 min irradiation. It should be noted that the irradiation-enhanced phosphorescence is dependent on not only newly formed cross-linked bond C-O-C, but also hydrogen bonding interactions. Upon increasing the irradiation time, the hydrogen bonding decreases and the newly formed C-O-C bond increases. In our manuscript, there are three steps in the irradiation process (Figure R17). At first, a large number of hydrogen bonding interactions formed in the doped systems suppresses intermolecular vibrations, so that the doped film state shows prolonged phosphorescence emission than that of the powder state (Figure 4a). Secondly, new covalent bond (C-O-C) and complex cross-linking network are formed in the PVA matrix after the irradiation, further suppressing the nonradiative transitions to enhance the phosphorescence in the system directly (Figure 4b). Thirdly, there are the rearrangements of hydrogen bonding interactions and formed C-O-C bonds in the system (Figure 4c). The chemical shift of H atoms in the benzene ring recovering back to the original position is tightly related with the rearrangement of hydrogen bonding interactions. This is a key step, because hydroxyl groups in PVA chain and in SDP are almost completely consumed by forming the cross-linked bonds (C-O-C) when the irradiation time exceeds 80 min, where the former formed hydrogen bonding would be broken to release free hydroxyl groups, and subsequently, these free hydroxyl groups continue to form new cross-linked

bonds, leading to the recovery of the chemical shift of H atoms in the benzene ring. In addition, the fact that films could nearly recover to their original status after 120 min of irradiation was also proven in Figure R6.

Figure R17 (Figure 4). Mechanism of IR RTP at room temperature. (a) Substantial hydrogen bonding interactions between SDP and SDP, SDP and PVA, and PVA and PVA in unirradiated SDP-based film. (b) Covalent bond (C-O-C) formation between PVA chains by 254 nm UV light irradiation for less than 45 min. (c) Rearrangement of hydrogen bonding interactions, and further formation of covalent bond (C-O-C) after the irradiation for more than 45 min. (d) ^1H NMR spectra of SDP-based film (1 mg/mL) in DMSO-d_6 under different irradiation times. (e) Electron paramagnetic resonance spectra and (f) one-dimensional scattering profiles in the q_z direction of GiWAXS pattern for SDP-based film (0.3 mg/mL) under different irradiation times.

Comment 11: In Fig. 2a, the phosphorescence lifetime of ABP doped film decreased from 26.59 ms to 25.06 ms, after 45 min irradiation, which showed very significant contrast compared with other doped systems. Corresponding explanation is needed.

Response: We thank the reviewer for this good suggestion. As shown in Figure R18a,c (Figure 2a,c), the phosphorescence lifetime (without irradiation: 26.59 ms; with irradiation 45 min: 25.06 ms) of ABP doped film is nearly unchanged before and after 45 min UV irradiation. Meanwhile, ABP-doped film has no afterglow emission before and after the irradiation. Compared with other phosphors in Figure R1, the two methyl groups in ABP may result in less hydrogen bonding formations than the sulfone group in SDP. At the same time

in Figure R18c (Figure 2c), the lifetime of ABP at powder state, doped film state without irradiation, and doped film state with 45 min irradiation is 2.13 ms, 26.59 ms, and 25.06 ms, respectively. As we discussed in the previous version (page 15), “Even if the hydroxyl groups at both ends of ABP can form hydrogen bonding interactions with the PVA matrix, two methyl groups are relatively large. The irradiated system could not fully suppress nonradiative transition of ABP, and thus did not show irradiation-enhanced phosphorescence.” Furthermore, the two-dimensional GiWAXS of ABP film was also conducted. As shown in Figure R4 (Supplementary Fig. 34), the scattering band at around 4.39 Å indicates no obvious π - π stacking interaction formation upon increasing the irradiation time. This is a main reason that ABP film has no irradiation-enhanced phosphorescence emission.

Related discussions have been added into the revised manuscript (page 17): “Meanwhile, the scattering band at around 4.39 Å from two-dimensional GiWAXS pattern (Supplementary Fig. 34) of ABP-doped film indicates no obvious π - π stacking interaction formation upon increasing the irradiation time. This is a main reason that ABP-doped film has no irradiation-enhanced phosphorescence emission.”

Figure R18 (Figure 2). Irradiation-responsive room-temperature phosphorescence of the eight hybrid films. (a) Photographs of the eight polymeric systems before and after turning off UV 254 nm light source. (b) IRRTP lifetime of SDP-based polymeric system at different doping concentrations by 254 nm light irradiation for 45 min. (c) Comparison of room

temperature phosphorescence lifetime of eight polymeric systems at different states. Red bar: phosphor powder without the PVA matrix. Green bar: phosphor-doped PVA films without the light irradiation. Blue bar: phosphor-doped PVA films with UV irradiation for 45 min.

Response to Reviewer #3's Comments:

In this paper, the authors reported the intriguing phenomenon of organic room temperature phosphorescent materials, i.e. light-induced phosphorescence enhancement. By various experiments, authors claimed that this phenomenon originates from the UV-induced covalent bond formation (e.g. C-O-C) of PVA matrix polymers which greatly reduces non-radiative decay processes of the phosphors.

While authors argued that this is the first report for light-induced phosphorescence enhancement, two reports for a similar phenomenon have already been published as follows, which greatly reduce the novelty of this work:

- 1) S. Reineke and coworkers, *Sci. Adv.* 7310 (2019)
- 2) J. Kim and coworkers, *Adv. Opt. Mater.* 2000654 (2020)

In those papers, authors claimed that the origin of the phenomenon stems from the conversion of triplet oxygen to singlet oxygen by UV-irradiation, leading to the unique phosphorescence enhancement of organic phosphors, which definitely challenges arguments in the current manuscript.

This reviewer believes that the experiments conducted in the current work would not fully rule out the possibility of the hypothesis suggested by previous works (i.e. UV-induced oxygen consumption and hence phosphorescence enhancement in the polymer matrix).

Considering those factors, this reviewer cannot support this work to be published in a premier journal such as *Nat. Commun.*

Response: Many thanks for your meaningful comments! First of all, to avoid any inappropriate argument about irradiation-enhanced green and yellow phosphorescence, the related statements have been modified in the revised manuscript.

The novelty of the irradiation-enhanced phosphorescence in this work

Because of the unclear mechanism of effective ultralong room temperature phosphorescence, and lack of common molecular design strategy, developing pure organic RTP materials under ambient conditions is highly important, especially for the systems with multiple enhanced phosphorescence by UV irradiation. Since the irradiation-dependent ultralong polymeric room temperature phosphorescence is very intriguing, and may have wide application potential in ink-free screen-printing technology, bioimaging, organic electronics, as well as information storage and security encryption, exploring more irradiation-dependent room temperature phosphorescent materials is quite desired.

In this work, a series of phosphor-doped PVA films with irradiation-enhanced phosphorescence has been found after screening many kinds of potential phosphors in our lab, indicating that the irradiation-enhanced phosphorescence originates from those newly cross-linked covalent bonds and hydrogen bonding interactions. This work is not only expected to eliminate the instability in light-responsive organic room temperature

phosphorescence systems under UV irradiation, but also provides a general design principle to develop irradiation-stimulating ultralong phosphorescence using amorphous polymers under ambient conditions.

Firstly, longer phosphorescence lifetime of irradiation-enhanced phosphorescence emission was achieved. Our work is different from previous work (*Sci. Adv.* 2019, 5, eaau7310 and *Adv. Opt. Mater.* 2020, 8, 2000654) that the phosphorescence lifetime of light-induced RTP materials was very short. After carefully checking these two publications, we found that both of two systems were irradiated under UV 365 nm, light-induced phosphorescence was from the oxygen consumption in the PMMA matrix, and the longest phosphorescence lifetime of 406 ms and 0.90 ms was obtained after UV exposure, respectively. Differently in our work, the irradiation-enhanced room temperature phosphorescence systems exhibit longer phosphorescence lifetime and afterglow, reaching to 828.81 ms and 8 s under ambient conditions, respectively. Most importantly, the mechanism of irradiation-enhanced phosphorescence in this work is ascribed to the cross-linked bands (C-O-C), hydrogen bonding, and π - π stacking interactions, rather than oxygen consumption.

Secondly, stable phosphorescence emission after irradiation was achieved in our work. (1) As we discussed in the "Introduction" section, light is an attractive stimulus for constructing responsive nanosystems, and UV light-responsive polymeric nanomedicine has received much attention for their applications in the spatial, temporal and on-demand drug delivery or disease therapy. However, a significant inherent challenge in most organic stimulus-responsive systems is their poor long-term stability and durability. Although substantial progress has been made in the field, developing more stable and advanced responsive systems for a wide range of applications is still highly desired. This character is expected to eliminate the instability in light-responsive systems under light irradiation. Therefore, the exploration of stimulus-responsive polymer-based room temperature phosphorescence materials is attractive and important. (2) Our phosphors show irreversible irradiation-enhanced phosphorescence as compared with *N,N'*-di(1-naphthyl)-*N,N'*-diphenyl-(1,1'-biphenyl)-4,4'-diamine (NPB) and Br6A doped PMMA systems in these two papers (*Sci. Adv.* 2019, 5, eaau7310 and *Adv. Opt. Mater.* 2020, 8, 2000654). Because the emitting layer undergoes an oxygen refilling process in the matrix, the light-induced phosphorescence would disappear in a small period of time. Even though the reversible phosphorescence in these papers is high important for repeatable noncontact labeling and the visualization of concealed information, it may still lack of the stability. Different from the reversible phosphorescence, our SDP-based phosphors show high stability and irreversible irradiation-enhanced phosphorescence after exposure to surrounding oxygen-included conditions.

Thirdly, we report a strategy to achieve ultralong phosphorescence. Different from the oxygen-consumption strategy to achieve irradiation-enhanced phosphorescence, in our

work, the formed cross-linked covalent bonds (C-O-C) are attributed to significantly suppress the nonradiative transitions. Lots of C-O-C covalent bonds were formed between PVA-PVA chains and phosphor-PVA after UV irradiation. A common strategy was proposed since irradiation-enhanced phosphorescence was also achieved in other hydroxyl group-containing polymer matrices. This work not only lays a foundation for the fabrication irradiation-responsive ultralong room-temperature phosphorescence (IRRTP) systems, but also paves a way for the future development in the field.

Lastly, based on the IRRTP feature of the developed doped polymer systems, we took one step further to demonstrate their potential application for polychromatic screen printing. Various patterns including pandas and lotuses were fabricated through a straightforward screen-printing technique by using eight phosphor-doped PVA. As a representative, SDP and 2,2-DB based films exhibit obvious green and yellow phosphorescence emission, respectively. Furthermore, an application in multilevel information encryption was also explored, and the phosphorescence emission color and brightness of these patterned letters would change upon increasing the irradiation time on the paper substrates at the same time. It is very rare that this afterglow color changes with increasing the irradiation time. This unique photophysical property enables potential applications in information storage, and multilevel anticounterfeiting without complicated fabrication processes.

As such, this work possesses high novelty to warrant its publication.

To exclude the influence of oxygen consumption

As the reviewer indicated, the previous papers (*Sci. Adv.* 2019, 5, eaau7310 and *Adv. Opt. Mater.* 2020, 8, 2000654) reported the light-induced phosphorescence. In these two papers, the triplet state of phosphor energy transfers to triplet state of oxygen, and then the triplet oxygen (T_n) generates the singlet oxygen (S_n). Both of them show the light-induced phosphorescence because of the consumption of oxygen in PMMA systems. Thanks to reviewer's kind suggestion, the two importance references have been cited in the revised manuscript (Reference No. 45 and 46).

We have confirmed that the irradiation-enhanced phosphorescence in our work is not caused by oxygen consumption. Firstly, as shown from **Figure R19** (Supplementary Fig. 28), there is no phosphorescence and irradiation-enhanced phosphorescence emission before and after 45 min irradiation for SDP doped PMMA, while SDP-doped PVA-100, PVA-87, and PVA-80 matrices show obvious irradiation-enhanced phosphorescence emission after UV irradiation. At the same time, SDP-doped PVA-co-PE also exhibits an obvious irradiation-enhanced phosphorescence emission (PVA-co-PE contains multiple hydroxyl groups). As we demonstrated in the manuscript, the intriguing irradiation-enhanced phosphorescence is resulted from the suppressed nonradiative transition by hydrogen bonding and cross-linked covalent bond (C-O-C) formations. Meanwhile, the SDP-doped

PVAc does not show irradiation-enhanced phosphorescence due to the absence of the hydroxyl groups, and thus they cannot form C-O-C covalent bonds after UV irradiation. Secondly, as seen from Figure R19a (Supplementary Fig. 28a), compared with the delayed phosphorescence spectra of SDP-doped PVA film, SDP-doped PMMA film shows very weak photoluminescence and delayed phosphorescence emission intensity. Meanwhile, in Figure R19b (Supplementary Fig. 28b), the emission spectra of SDP-doped PMMA show a little change before and after 45 min irradiation at ambient conditions, indicating that SDP-doped PMMA without irradiation-enhanced phosphorescence emission is not caused by the oxygen presence.

To give more insights, we have carried out the UV irradiation experiments in a glove box, in which the oxygen level is maintained below 0.01 ppm. In Figure R20a (Supplementary Fig. 29a), obvious emission enhancement was observed after 45 min continuous UV irradiation, and the phosphorescence emission of the irradiated and unirradiated films was distinguishable. In addition, an ethylene-vinyl alcohol copolymer-sealed oxygen-barrier layer (PVA-co-PE) was deposited on top of the sample to avoid the exposure of the emitting layer to surrounding oxygen (sealed in the glove box to exclude oxygen entrapping). As shown in Figure R20b,c (Supplementary Fig. 29b,c), phosphorescence emission photographs were recorded in the cuvette that was filled with oxygen gas. Before the irradiation, the SDP-doped PVA film and SDP-doped PVA film with O₂ barrier have no visible phosphorescence. After 45 min irradiation, both films show obvious green afterglow emission and the duration time is about 9 s. The results confirm that the luminescence changing behavior is not obviously related to the existence of oxygen. To further confirm that the phosphorescence enhancement is not related to the oxygen level in the polymer matrix, the phosphorescence emission spectra and phosphorescence lifetime decay curves have been measured at different surrounding conditions. In Figure R21, with a main 488 nm emission peak, SDP-based system has almost same phosphorescence emission intensity at different surrounding conditions. Meanwhile, the phosphorescence decay curves also show a similar trend of lifetime in vacuum, nitrogen, and air atmospheres. Because of the high susceptibility of triplet excitons to oxygen, the phosphorescence lifetime of SDP-doped film in oxygen condition is slightly lower than that in vacuum and nitrogen.

To sum up, SDP-based film does not show phosphorescence emission before the irradiation, and the irradiation-enhanced phosphorescence is resulted from the suppressed nonradiative transition by the formed hydrogen bonding and cross-linked covalent bonds (C-O-C) under the irradiation. To examine the effect of oxygen on the irradiation-dependent phosphorescence, the photophysical properties of different SDP-doped matrices (PVA, PVA-87%, PVA-80%, PVA-co-PE, PVAc, and PMMA) and SDP-doped PVA in different conditions (i.e., vacuum, nitrogen, oxygen, air) clearly indicate that the phosphorescence enhancement is not obviously related to the oxygen level.

Figure R19 (Supplementary Fig. 28). Photophysical properties of SDP-doped films at ambient conditions. (a) Delayed photoluminescence spectra of SDP-doped with PVA and PMMA matrix. (b) Normalized prompt photoluminescence spectra of SDP-doped PMMA (delayed time is 5 ms).

Figure R20 (Supplementary Fig. 29). Photographs of SDP-doped systems before and after turning off UV 254 nm light source. (a) Phosphorescence emission of SDP-doped film irradiated at ambient conditions and in a glove box. (b,c) Phosphorescence emission of SDP-doped film and SDP-doped film sealed with an ethylene-vinyl alcohol copolymer as an oxygen-barrier layer irradiated for 0 min and 45 min, respectively. Cuvette was filled with oxygen.

Figure R21. Photophysical properties of SDP-doped film at 0.3 mg / mL concentration. (a) Phosphorescence emission spectra at different surrounding conditions. (b) Decay curves of phosphorescence lifetime at different conditions.

REVIEWERS' COMMENTS

Reviewer #2 (Remarks to the Author):

This revision could be accepted for publication.

Reviewer #3 (Remarks to the Author):

The authors' original data along with additional experiments and explanations confirm the formation of crosslinking under continuous irradiation of UV light. Also, this phenomenon is different from the previously reported UV-induced phosphorescence enhancement. Thus, this reviewer could support the publication of this work.

REVIEWER COMMENTS

Reviewer #2 (Remarks to the Author):

This revision could be accepted for publication.

Response: Many thanks for your kind recommendation.

Reviewer #3 (Remarks to the Author):

The authors' original data along with additional experiments and explanations confirm the formation of crosslinking under continuous irradiation of UV light. Also, this phenomenon is different from the previously reported UV-induced phosphorescence enhancement. Thus, this reviewer could support the publication of this work.

Response: We appreciate your recommendation of publication.